# An extracellular matrix protein promotes anillin-dependent processes in the *Caenorhabditis elegans* germline

Hongxia Lan[1,4], Xinyan Wang[1,2,3], Ling Jiang[1,2], Jianjian Wu[1], Xuan Wan[1,2], Lidan Zeng[1], Dandan Zhang[1,2], Yiyan Lin[1], Chunhui Hou[1] , Shian Wu[4], Yu Chung Tse[1,2]

**Cell division requires constriction of an actomyosin ring to segregate the genetic material equally into two daughter cells. The spatial and temporal regulation of the contractile ring at the division plane primarily depends on intracellular signals mediated by the centralspindlin complex and astral microtubules. Although much investigative work has elucidated intracellular factors and mechanisms controlling this process, the extracellular regulation of cytokinesis remains unclear. Thus far, the extracellular matrix protein Hemicentin (HIM-4) has been proposed to be required for cleavage furrow stabilization. The underlying molecular mechanism, however, has remained largely unknown. Here, we show that HIM-4 and anillin (ANI-1) genetically act in the same pathway to maintain the rachis bridge stability in the germline. Our FRAP experiments further reveal that HIM-4 restricts the motility of ANI-1. In addition, we demonstrate that HIM-4 is recruited to the cleavage site in dividing germ cells and promotes the proper ingression of the cleavage membrane. Collectively, we propose that HIM-4 is an extracellular factor that regulates ANI-1 for germ cell membrane stabilization and contractile ring formation in *Caenorhabditis elegans* germline cells.**

## Introduction

The ECM is a tissue-specific assembly of molecules that reside and function outside of the cell. Specific resident cells secrete these molecules, mainly proteoglycans and large, multidomain, fibrous proteins, which form extracellular fibrils and supramolecular networks (Keeley & Mecham, 2013). ECM proteins provide structural support for cells and tissues (Frantz et al, 2010). They also regulate cell determination, differentiation, proliferation, polarity, and migration (Hynes, 2009; Frantz et al, 2010).

Apart from the roles in tissue organization, some of the ECM proteins are also involved in cell division. Previous work has shown that chondroitin proteoglycans (CPGs) are required for *Caenorhabditis elegans* (*C. elegans*) embryo cell division (Mizuguchi et al, 2003; Olson et al, 2006; Izumikawa et al, 2010). RNA interference (RNAi)–mediated simultaneous depletion of CPG-1 and CPG-2, two of the nine CPG proteins in *C. elegans*, caused defective cell division (Olson et al, 2006). In *cpg-1/cpg-2* double RNAi zygotes, chromosome segregation proceeded normally, but the cleavage furrow failed to form during anaphase, resulting in multinucleated single-cell embryos (Olson et al, 2006). However, this defect may be caused by the imbalanced osmotic pressure in *cpg-1/cpg-2 (RNAi)* zygotes.

Recently, another extracellular matrix protein, Hemicentin (HIM-4), has been proposed to be required for germline syncytium stabilization. Depletion of HIM-4 resulted in effects on the germ cell, including membrane destabilization, cleavage furrow retraction, and cytokinesis failure, resulting in multinucleated cells in the germline (Xu and Vogel, 2011a, 2011b; Vogel et al, 2011). Similarly, knockdown or targeted inactivation of Hemicentin-1 in mouse embryos also caused membrane destabilization, cleavage furrow retraction, and cytokinesis failure, which resulted in a large number of embryos arrested at the one- to four-cell stage (Xu & Vogel, 2011b). These results indicate that HIM-4 is required for proper cytokinesis, perhaps with a direct role. However, the molecular mechanism by which cytokinesis is regulated is not yet known.

Hemicentins are a highly conserved class of ECM proteins within metazoans and contain multiple domains, including a conserved von Willebrand A domain, a long chain of immunoglobulin modules, a series of EGF-like modules, and a carboxyl-terminal fibulin-type module (Whittaker & Hynes, 2002; Argraves et al, 2003; Dong et al, 2006). Hemicentins were first identified in *C. elegans*, which contain

[1]Department of Biology, Southern University of Science and Technology (SUSTech), Shenzhen, China  [2]Guangdong Provincial Key Laboratory of Cell Microenvironment and Disease Research, Shenzhen Key Laboratory of Cell Microenvironment, SUSTech, Shenzhen, China  [3]Centre of Reproduction, Development and Aging, Faculty of Health Sciences, University of Macau, Macau, China  [4]State Key Laboratory of Medicinal Chemical Biology, College of Life Sciences, Nankai University, Tianjin, P.R. China

Correspondence: tseyc@sustech.edu.cn
Hongxia Lan, Xinyan Wang, and Ling Jiang are co-first authors

a single representative called HIM-4 (Hodgkin et al, 1979). Subsequently, two vertebrate paralogues, Hemicentin-1 and Hemicentin-2, were identified (Schultz & Onfelt, 2001; Vogel et al, 2006). HIM-4 is secreted by skeletal muscles and gonad leader cells. It is delivered to the extracellular space in the gonad to stabilize the germline syncytium (Vogel & Hedgecock, 2001). In the *C. elegans* gonad, HIM-4 forms quasi-hexagonal lattice tracks in the mitotic region, and a diffuse sheet surrounding the rachis (Vogel & Hedgecock, 2001). Mutation of the *him-4* locus and depletion of HIM-4 result in a high incidence of male offspring, defective germ cell migration, and chromosome instability (Hodgkin et al, 1979; Vogel & Hedgecock, 2001). Previous evidence in mouse and zebra fish revealed that Hemicentin has pleiotropic functions in transient and stable cell contacts because of its involvement in maintaining the architectural integrity and tensile strength of tissues and organs (Carney et al, 2010; Feitosa et al, 2012). Similar tissue instability is also reported in human macular disease, in which patients carrying a polymorphism in human Hemicentin-1 would suffer from macular degeneration with the onset of this disease being age-dependent (Schultz et al, 2003; Thompson et al, 2007). This indicates that Hemicentin not only plays a scaffolding role within tissues of lower organisms but is essential for human health, particular to the elderly.

In this study, using *C. elegans* germline as a model system, we show that HIM-4 localizes to the rachis bridge and the cleavage plane of dividing germ cells, and this localization is necessary to recruit anillin (ANI-1). Simultaneously depletion of ANI-1 and HIM-4 phenocopies the single depletion of each for germline compartmentation, rachis bridge size, and cleavage furrow constriction rate of the dividing germ cells. This suggests that HIM-4 regulates ANI-1 for membrane stabilization and contractile ring formation. Our work, therefore, demonstrates that the ECM protein HIM-4 mediates two biological events: (1) stabilization of germ cell membrane and (2) promotion of contractile ring constriction.

## Results

### HIM-4 promotes germline compartmentation and intercellular bridge stability

Extracellular matrix protein HIM-4 is enriched within the *C. elegans* gonad rachis to stabilize syncytial architecture (Dong et al, 2006; Xu & Vogel, 2011b) (Fig 1A and B). It is also required to prevent the multinucleation of germ cells (Dong et al, 2006; Xu & Vogel, 2011b). For this study, we made extensive use of the *him-4(e1266)* (Fig S1) mutant line that was originally characterized to produce a high incidence of males (Him) (Hodgkin et al, 1979). In the control hermaphrodite gonad expressing GFP::PH and mCherry::HIS (control in this study refers to the worms fed with HT115 bacteria expressing

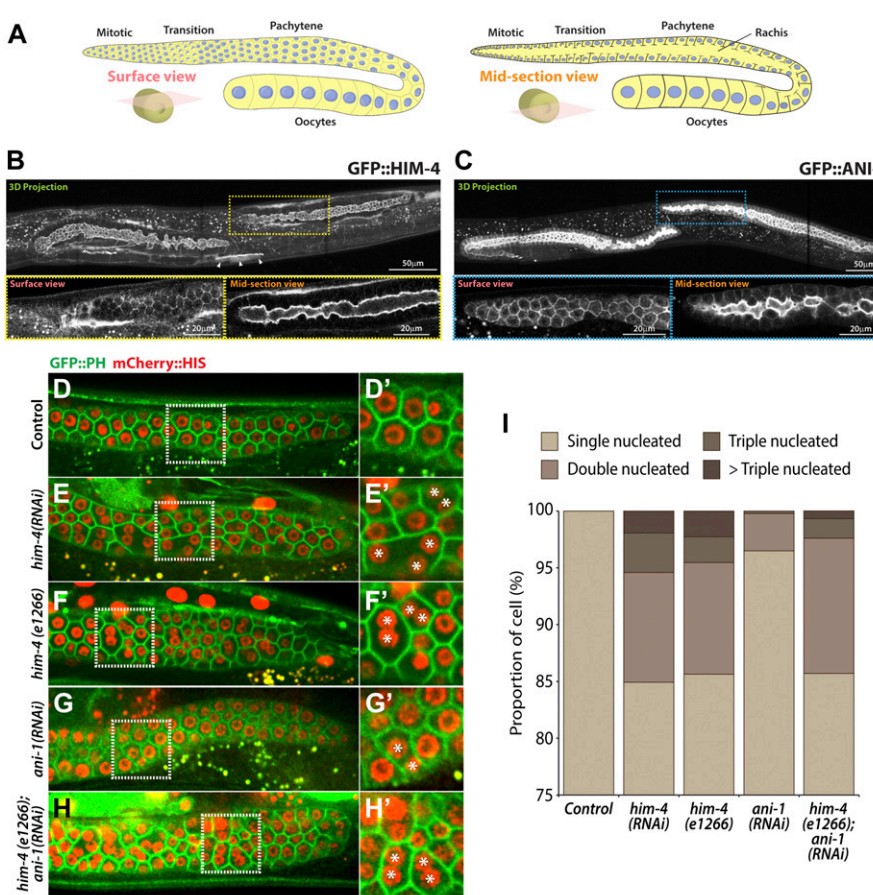

Figure 1. Multinucleated germ cell formation and subcellular localization of HIM-4 and ANI-1 in the *C. elegans* germline.
**(A)** Schematic representation of the adult hermaphrodite germline. **(B)** 3D projection, surface view, and midsection view of the hermaphrodite germline expressing GFP::HIM-4. Scale bar, 20 and 50 μm. **(C)** 3D projection, surface view, and midsection view of the hermaphrodite germline expressing GFP::ANI-1. Scale bar, 20 and 50 μm. **(D–H)** Representative confocal images of germ cells expressing GFP::PH and mCherry::HIS in the control (D), *him-4 (RNAi)* (E), *him-4(e1266)* (F), *ani-1(RNAi)* (G), and *him-4(e1266); ani-1(RNAi)* (H). Scale bar, 20 μm. **(D'–I')** Magnified image from the boxed region. **(I)** Quantification of the number of multinucleated germ cells in control, *him-4(RNAi)*, *him-4(e1266)*, *ani-1(RNAi)* and *him-4(e1266);ani-1(RNAi)* germline cells. In each condition, 250 to 300 germ cells in 10 worms were counted.

empty vector L4440), each of the germ cells contains a single nucleus (Fig 1D), but 15% of the *him-4(RNAi)* and *him-4(e1266)* germ cells are multinucleated (Fig 1E, F, and I). To study how these multinucleated germ cells are formed, we imaged germ cell division by time-lapse confocal microscopy and traced the division process for an hour. Control germ cells and *him-4(RNAi)* germ cells started to divide in the mitotic zone, and the cleavage membranes remained stable (i.e., no membrane regression) during the imaging periods. On the contrary, about 10% of the *him-4(RNAi)*–treated dividing germ cells showed cleavage membrane retraction at a later stage of furrow ingression, resulting in binucleation (Fig 2A, cell ii and Video 1) (Dong et al, 2006; Xu & Vogel, 2011b). Membrane retraction and resultant binucleation were also observed in the meiotic transition zone cells (Fig 2A, cell iii and iv and Video 1). These results indicate that HIM-4 affects *C. elegans* germline compartmentation in both the mitotic and transition regions.

The size of the rachis bridge is another parameter that can be assessed for intercellular bridge stability within the *C. elegans* germline (Rehain-Bell et al, 2017). To determine if HIM-4 affects the integrity or formation of the rachis bridge, we imaged the pachytene gonad expressing GFP::PH from top to bottom. The largest diameter of the rachis bridge at the sagittal plane was measured (Fig 2B). Notably, the average diameter of rachis bridges of *him-4(e1266)* (4.57 ± 0.09 µm, n = 64) and *him-4(RNAi)* (3.97 ± 0.07 µm, n = 71) is significantly larger than that of control germ cells (3.01 ± 0.06 µm, n = 74) (Fig 2C). To ensure that we were viewing the correct diameter, we also quantified the data from the surface of the rachis bridge, which

are consistent with the analysis from the sagittal view (Fig S2A and B). Collectively, these results suggest that HIM-4 is required for membrane compartmentation and intercellular bridge stability in the germline.

## HIM-4 and ANI-1 act in the same pathway for intercellular bridge stability

In addition to HIM-4, the anillin actomyosin scaffold protein family members ANI-1 and ANI-2 also localized in the rachis (Amini et al, 2014) (Fig 1C), and both are required for rachis bridge organization (Amini et al, 2014; Zhou et al, 2013). Our data and that of another study have demonstrated that the rachis bridge size is reduced in *ani-2(RNAi)* but enlarged in *ani-1(RNAi)* gonads (Fig 2C) (Amini et al, 2014; Rehain-Bell et al, 2017). Interestingly, the rachis bridges of *ani-1(RNAi)* germline cells were enlarged to a very similar extent to that of *him-4*(RNAi) (Fig 2C). In addition, multinucleated germ cells were observed in *ani-1(RNAi)* (Fig 1G and I), and similar proportion of multinucleated germ cells as *him-4(RNAi)* and *him-4(e1266)* was observed in *ani-1(RNAi);him-4(e1266)* (Fig 1H and I). Therefore, we wondered whether HIM-4 and ANI-1 may act in the same pathway for rachis bridge organization. Towards addressing this question, we measured the pachytene rachis bridge diameter of the *him-4 (e1266)* line depleted of ANI-1 by RNAi. The maximum diameter of the rachis bridge in *him-4(e1266);ani-1(RNAi)* was significantly larger than the control but not significantly different from *him-4 (e1266)* (Fig 2C). This suggests that HIM-4 and ANI-1 act in the same

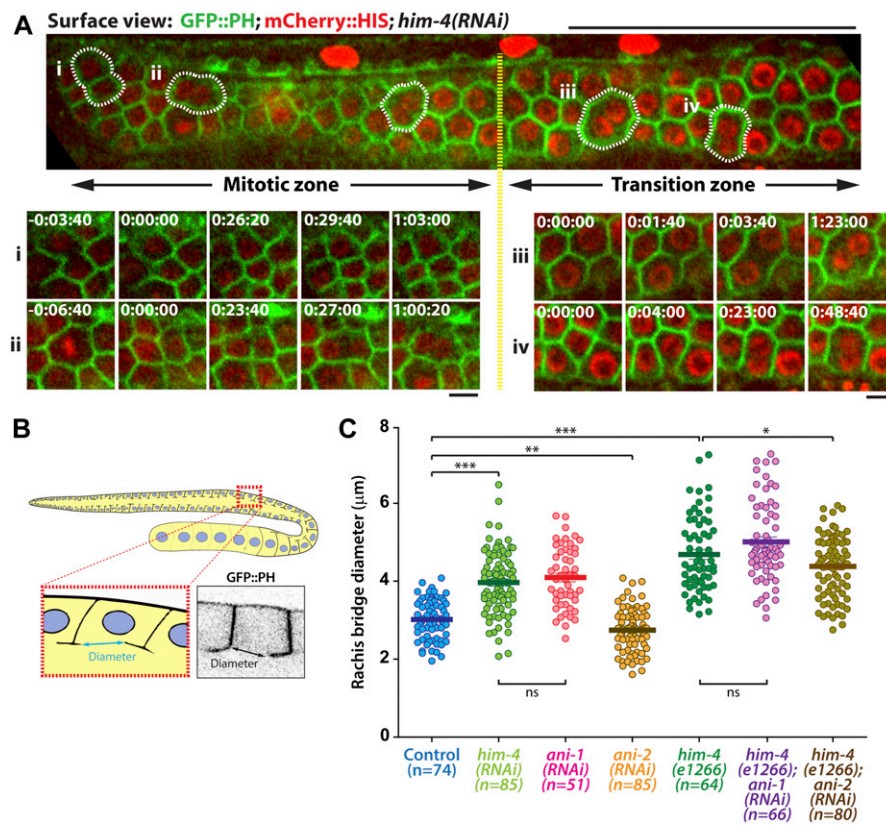

**Figure 2. Germline cell division and rachis bridge diameter in the *C. elegans* hermaphrodite germline.**
**(A)** Representative time-lapse confocal images of GFP::PH; mCherry::HIS; *him-4(RNAi)* expressing germline cells in mitotic and transition zones. Scale bar, 50 and 5 µm. **(B)** Schematic depiction of the rachis bridge in the germline cells expressing the membrane marker GFP::PH. **(C)** Quantification of germline cell rachis bridge diameter in control, *him-4(RNAi), him-4(e1266), ani-1(RNAi), ani-2(RNAi), him-4(e1266); ani-1(RNAi)* and *him-4(e1266); ani-2(RNAi)*. *P < 0.05 (*t* test); ***P < 0.001 (*t* test). Sample sizes are represented by *n*. ns, nonsignificant.

pathway for intercellular bridge stability in the germline. To corroborate this model, we depleted ANI-2, a putative competitor of ANI-1, in the *him-4(e1266)* mutant and then measured rachis bridge diameter. The enlargement of rachis bridge diameter in HIM-4 depletion would be resolved if it is caused by the reduction of ANI-1 activity. Our results show a statistically significant decrease in the average rachis bridge diameter of *him-4(e1266);ani-2(RNAi)* compared with that of *him-4(e1266)* (Fig 2C). This demonstrates that the intercellular bridge diameter in *him-4(e1266)* could be slightly rescued by the depletion of ANI-2. Taken together, our results demonstrate that ANI-1 and HIM-4 are required for intercellular bridge stability and probably act in the same pathway.

### HIM-4 stabilizes the localization of ANI-1 at rachis bridges

Given that HIM-4 acts in the same pathway with ANI-1 to promote intercellular bridge stability, we next sought to examine whether

HIM-4 regulates ANI-1. First, we tested whether HIM-4 promotes the recruitment of ANI-1. The overall intensity of GFP::ANI-1 at the rachis region in the control and *him-4(RNAi)* were compared, but no significant difference was observed (data not shown). We then examined whether HIM-4 regulates the mobility of ANI-1 by analyzing the diffusion behavior of HIM-4 and ANI-1 on the rachis membrane in the mitotic zone by fluorescence recovery after photobleaching (FRAP) (Fig 3A). HIM-4 showed an absence of recovery after photobleaching in the control germline at the rachis, which is consistent with HIM-4 as a cross-linked ECM protein (Fig 3B and Video 2). On the other hand, 50% of the GFP::ANI-1 intensity was recovered after photobleaching in the control (Fig 3C and Video 3). If HIM-4 confines ANI-1 at the rachis bridge, then ANI-1 should have greater mobility in the absence of HIM-4 and show greater recovery of fluorescence intensity. To test this, we depleted HIM-4 by RNAi in worms expressing GFP::ANI-1. The time to half-maximal recovery is significantly reduced from 100 ± 15.3 s in the control to 79.8 ± 12.6 s in

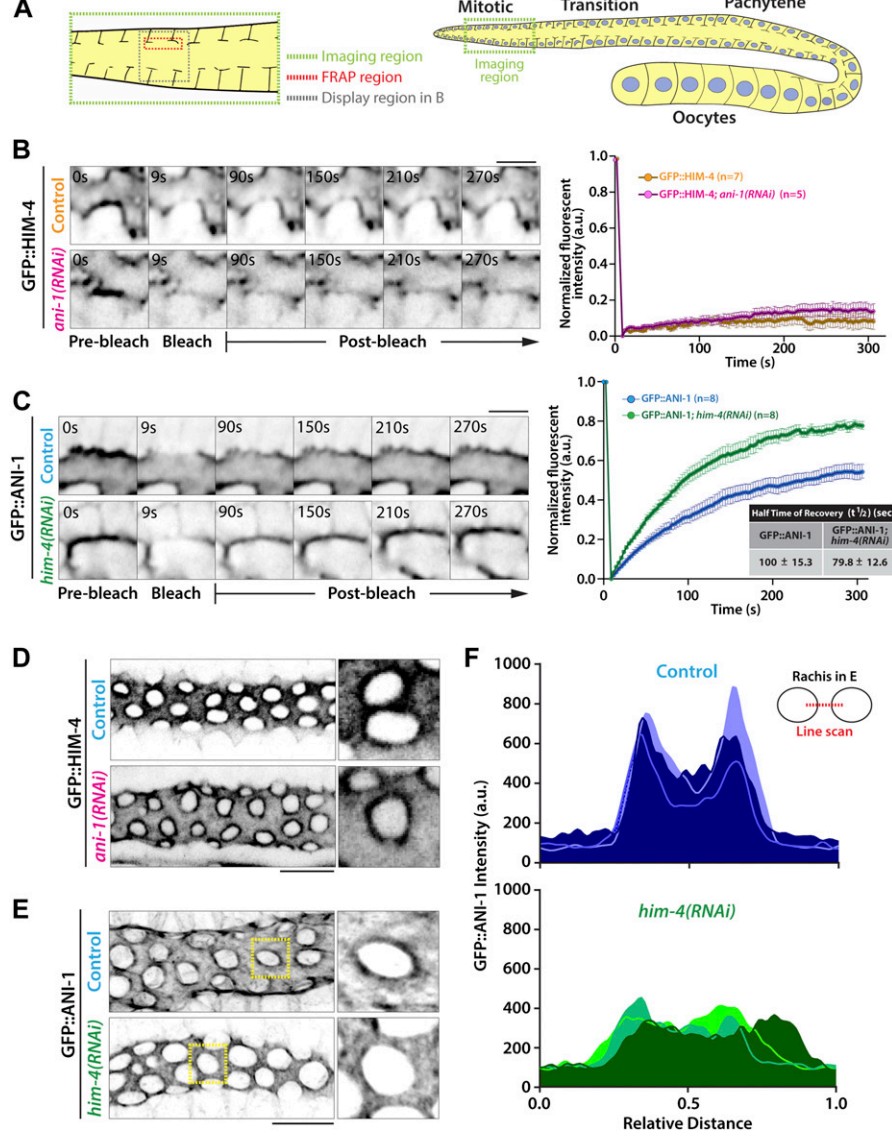

**Figure 3. HIM-4 regulates the mobility and localization of anillin in the *C. elegans* gonad.**
**(A)** Schematic depicting the region for FRAP experiments in the *C. elegans* germline. **(B)** Representative time-lapse confocal images and normalized recovery curves of GFP::HIM-4 and GFP::HIM-4; *ani-1(RNAi)* FRAP experiments. Scale bar, 5 μm. Error bars represent ± SEM; sample sizes are represented by *n*. **(C)** Representative time-lapse confocal images and normalized recovery curves of GFP::ANI-1 and GFP::ANI-1; *him-4(RNAi)* FRAP experiments. Scale bar, 5 μm. Error bars represent ± SEM; sample sizes are represented by *n*. **(D)** Representative confocal images of GFP::HIM-4 at the rachis bridges in control and ani-1(RNAi). Scale bar, 10 μm. **(E)** Representative confocal images of GFP::ANI-1 at the rachis bridges in control and him-4(RNAi). Scale bar, 10 μm. **(F)** Quantification of the GFP::ANI-1 intensity across the rachis bridges in control and *him(RNAi)*. Three representative data were shown in each condition.

*him-4(RNAi)* (Fig 3C and Video 4). This result was corroborated by the recovery analysis performed at the pachytene zone (Fig S3A and B). As a comparison, ANI-2 and non-muscle myosin II (NMY-2) have functions correlated to ANI-1 and are abundant in the rachis. However, they had variable recovery behaviors after photo-bleaching. The absence of recovery was observed in GFP::ANI-2 (Fig S3C and Video 5), whereas NMY-2::GFP was recovered slowly (Fig S3D and Video 6). Importantly, HIM-4 depletion did not alter the recovery rate of GFP::ANI-2 (Fig S3C and Video 7) and NMY-2::GFP (Fig S3D and Video 8) after photobleaching in the mitotic zone. Thus, our data indicate that HIM-4 inhibits the mobility of ANI-1.

The subcellular localization of ANI-1 was then examined. If HIM-4 regulates the localization of ANI-1, then the localization patterns of ANI-1 should be distinct in the wild-type versus HIM-4–depleted background. In *C. elegans* germlines, ANI-1 and HIM-4 accumulated in the rachis and formed ring-like structures at the rachis bridges (Fig 3D and E). The circular shape of the rachis bridge and the pattern of GFP::HIM-4 accumulating around the rachis bridges were

unaffected in ANI-1 depletion (Fig 3D). In contrast, the ring-like structure of GFP::ANI-1 at the rachis bridge was absent in *him-4 (RNAi)* (Fig 3E and F). Collectively, our results demonstrate that HIM-4 retains ANI-1 at the *C. elegans* rachis bridge.

## HIM-4 localizes to the division plane in *C. elegans* germ cells

HIM-4 regulates ANI-1 mobility and ANI-1 is a key component of the actomyosin contractile ring (Oegema et al, 2000; Piekny & Glotzer, 2008). We, therefore, considered whether HIM-4 could regulate cytokinesis through ANI-1. To test this, we imaged the subcellular localization of GFP::HIM-4 in dividing germ cells by spinning disk confocal microscopy, and then measured the GFP intensity at the division plane upon anaphase onset. In the first 100 s upon chromosome segregation, GFP::HIM-4 was slowly recruited to the division plane. After that, HIM-4 accumulated at the cleavage site and reached the maximum intensity at around 300 s after anaphase onset (Fig 4A and B). The detailed HIM-4 dynamics in germ cell

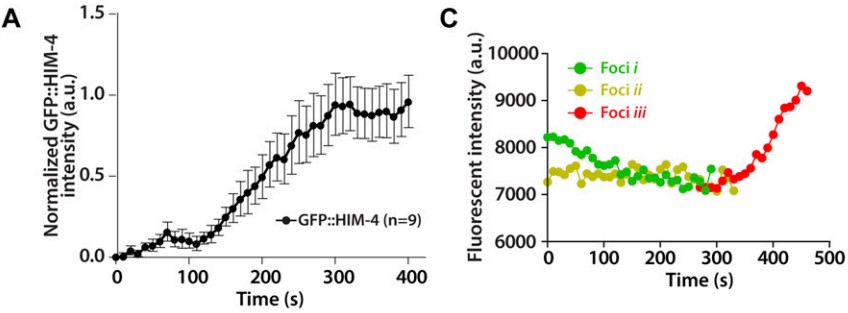

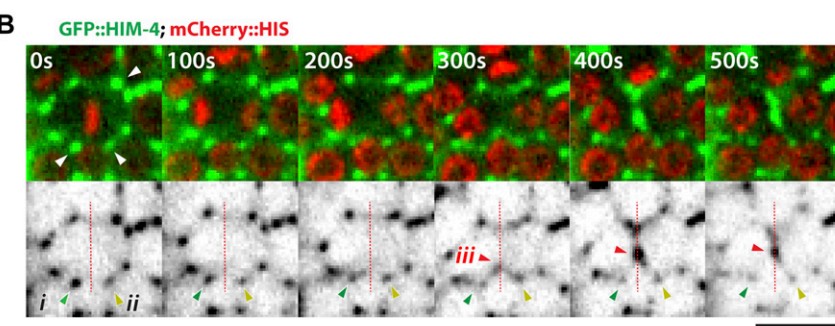

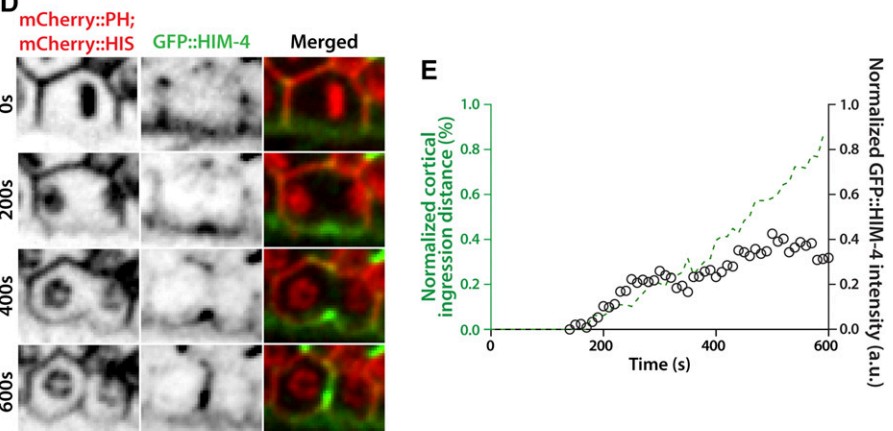

**Figure 4. HIM-4 localizes to the cleavage site in *C. elegans* dividing germline cells.**
**(A)** Quantification of GFP::HIM-4 at the cleavage site in *C. elegans* dividing germline cells. Sample sizes are represented by *n*. **(B)** Representative time-lapse confocal images of *C. elegans* germline cells expressing GFP::HIM-4; mCherry::HIS upon anaphase onset. The upper panels are merged green and red channels, and the lower panels show only the green channels displayed as inverted grayscale images. The white arrows in the upper panel indicate GFP::HIM-4 patches in the extracellular space. Foci *i* (green arrow) and foci *ii* (yellow arrow) are the two GFP::HIM-4 patches adjacent to the division plane. Foci *iii* (red arrow) indicates the newly formed patch in the division plane. Scale bar, 20 *μm*. **(C)** Quantification of fluorescence intensity for Foci *i*, *ii*, and *iii* in (B). **(D)** Representative time-lapse confocal images of germ cells expressing GFP::HIM-4; mCherry::HIS; mCherry::PH upon anaphase onset. Scale bar, 10 *μm*. **(E)** Quantification of cleavage furrow ingression and GFP::HIM-4 accumulation in dividing germ cells upon anaphase onset.

division was further examined by the time-lapse high-resolution confocal imaging. On the surface view of the mitotic zone germline expressing GFP::HIM-4, GFP-fluorescent patches were mainly present within the extracellular space between germ cells (Fig 4B). These patches were stable in terms of location and intensity. However, the status of these patches changed once their surrounding germ cells entered anaphase. Upon anaphase onset, the intensity of patches gradually decreased (Fig 4B and C, Foci *i* & *ii* and Video 9), and then part of each patch appeared to split with fragments moving towards the division plane and combining to form a new patch (Fig 4B, Foci *iii*). The intensity of this newly formed patch then gradually increased and moved with the ingressing cleavage furrow (Fig 4B and C). We also imaged the dividing germ cells co-expressing mCherry::PH; mCherry::HIS; GFP::HIM-4 from anaphase onset until the completion of furrow ingression (Fig 4D and Video 10). Concurrent with the initiation of membrane ingression, GFP::HIM-4 started to accumulate at the division plane.

The rates of GFP::HIM-4 accumulation and membrane ingression were closely correlated from 150 to 250 s after anaphase onset. Afterwards, the accumulation slowed down, and the patch fluorescent intensity remained stable until the ingression stopped (Fig 4E). Taken together, these data indicate that HIM-4 localizes to the division plane and moves with the cleavage furrow during anaphase in the dividing germ cells.

### HIM-4 regulates the constriction of the contractile ring

HIM-4 localization to the division plane indicates that it is likely to be involved in germ cell cytokinesis. We examined the cleavage furrow membrane ingression dynamics by time-lapse spinning disk confocal imaging in germ cells expressing NMY-2::GFP and mCherry:: HIS. For control germ cells, the average rate of ingression from anaphase onset to 80% membrane ingression was 482 s (Figs 5A and S4A and Video 11), whereas *him-4(RNAi)* and *him-4(e1266)*

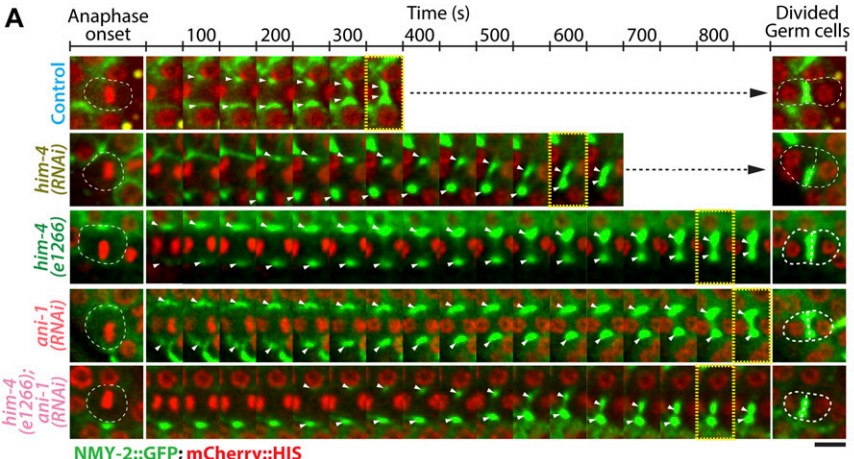

**Figure 5. HIM-4 and ANI-1 regulate the constriction of the contractile ring.**
**(A)** Representative time-lapse confocal images of germline cells expressing NMY-2::GFP; mCherry::HIS in the control, *him-4(RNAi)*, *him-4(e1266)*, *ani-1(RNAi)*, and *him-4(e1266);ani-1(RNAi)* upon anaphase onset. Images were recorded at 30-s intervals. Yellow boxes indicate approximate 10% cleavage furrow ingression. Scale bar, 10 μm. **(B)** Quantification of normalized cortical distance upon anaphase onset in the control, *him-4(RNAi)*, *him-4 (e1266)*, *ani-1(RNAi)*, and *him-4(e1266);ani-1(RNAi)* germline cells. **(C)** Quantification of the time required to reach 10% membrane ingression in the indicated conditions. Error bars represent ± SEM; **$P < 0.01$ ($t$ test); ***$P < 0.001$ ($t$ test). **(D)** Quantification of the constriction rate from anaphase onset to 10% membrane ingression in the conditions as indicated. Error bars represent ± SEM. **$P < 0.01$ ($t$ test); ***$P < 0.001$ ($t$ test).

dividing germ cells were delayed to 694 and 799 s, respectively (Figs 5A and S4A and Video 12). Chromosome segregation and contractile ring constriction are major events that affect the timing of the completion of cleavage furrow ingression. We analyzed chromosome dynamics by measuring the distance of chromosome segregation during anaphase, but no significant difference in chromosomes dynamics was observed in the control versus *him-4 (RNAi)* dividing germ cells (Fig S4C). This indicates that HIM-4 does not regulate chromosome segregation. On the other hand, we observed that the ingression of germ cell cleavage furrows followed two phases of constriction: a slow phase during the early anaphase and a fast phase during mid-anaphase until 80% membrane ingression (Fig 5B). The transition from slow phase to fast phase in the control dividing germ cells was quicker than *him-4(RNAi)* and *him-4 (e1266)* (Fig 5B). Because control cells transition to the fast phase on average at 10% membrane ingression, we set this as the endpoint of our measurement to compare the different conditions. As shown in Fig 5C, control germ cells took 110 s, whereas *him-4(RNAi)* and *him-4 (e1266)* required 190 s to reach 10% membrane ingression. The constriction rate to 10% membrane ingression was $6.2 \times 10^{-4}$ $\mu$m/s in the control germ cells, whereas it was significantly reduced to $3.57 \times 10^{-4}$ $\mu$m/s and $4.23 \times 10^{-4}$ $\mu$m/s in *him-4(RNAi)* and *him-4 (e1266)* germ cells, respectively (Fig 5B' and D).

### HIM-4 promotes ANI-1 recruitment to regulate the constriction of contractile ring in dividing germ cells

Our data demonstrate that HIM-4 localizes to the presumptive cleavage furrow and acts in the same pathway with ANI-1 to stabilize germline cell membrane. We, therefore, reasoned that HIM-4 may promote the recruitment of ANI-1 to the equatorial cortex and in turn manipulate the membrane ingression. To test this, we first measured ANI-1 accumulation at the equatorial region by imaging and analyzing the GFP::ANI-1 fluorescence intensity changes from anaphase onset (Fig 6A). In control dividing germ cells expressing GFP::ANI-1, the fluorescence intensity at the division plane increased by 50% in 145 ± 12.8 s, whereas it took 198 ± 13.3 s in *him-4 (RNAi)* germ cells (Fig 6B). In contrast, ANI-1 depletion did not compromise the accumulation rate of GFP::HIM-4 at the division plane (Fig S5). In addition, *him-4(RNAi)* germ cells took longer time to recruit ANI-1 to the same level as the control for similar extend of ingression in the early anaphase (Fig 6C). Altogether, these results demonstrate that HIM-4 is necessary for the recruitment of ANI-1 at the equatorial region of the cell.

To further test our hypothesis that HIM-4 regulates the formation of the germline cell contractile ring through ANI-1 recruitment, the dividing *ani-1(RNAi)* germ cells co-expressing NMY-2::GFP and mCherry:: HIS were imaged and the rate of membrane ingression was quantified (Fig 5A and Video 13). As observed for *him-4(RNAi)* and *him-4(e1266)* dividing germ cells, *ani-1(RNAi)* dividing germ cells took longer to reach 80% membrane ingression compared with the control germ cells (Figs 5B and S4A), and the time for constriction rate to 10% membrane ingression ($2.74 \times 10^{-4}$ $\mu$m/s) was significantly lesser than the control germ cells (Fig 5D). Significantly, *ani-1(RNAi)* dividing germ cells had no differences in measured constriction rate characteristics compared with *him-4(RNAi)* and *him-4(e1266)* dividing germ cells (Fig 5). Perhaps, even more importantly, the membrane ingression behaviors of *him-4 (e1266);ani-1(RNAi)* dividing germ cells have no significant differences from *him-4(RNAi)*, *him-4(e1266)*, and *ani-1(RNAi)* (Figs 5 and S4A, and B and Video 14). Collectively, these data indicate that HIM-4 is necessary to recruit ANI-1 to the presumptive cleavage furrow position for proper contractile ring constriction in *C. elegans* germ cells.

## Discussion

In this article, we have expanded our understanding of the role played by the extracellular matrix protein Hemicentin beyond the

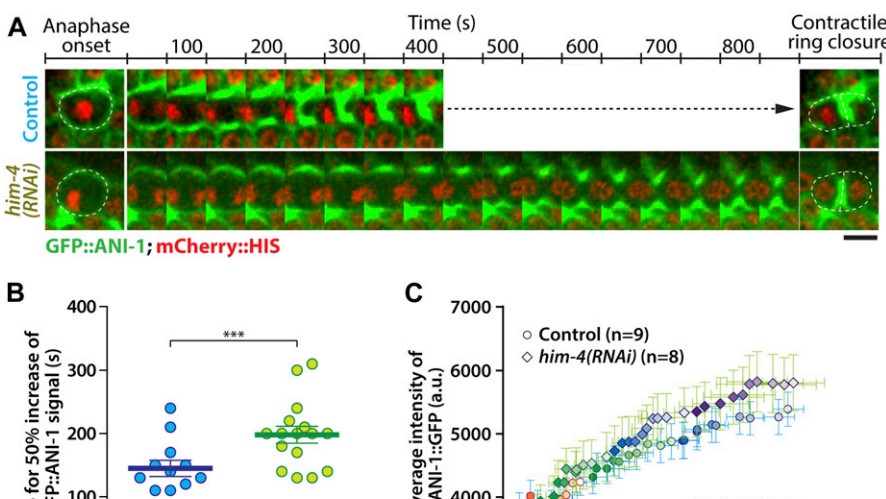

**Figure 6. HIM-4 promotes ANI-1 accumulation at the cleavage site of dividing germline cells.**
**(A)** Representative time-lapse confocal images of germline cells expressing GFP::ANI-1; mCherry::HIS in the control and *him-4(RNAi)* upon anaphase onset. Scale bar, 10 $\mu$m. **(B)** Amount of time necessary to increase GFP::ANI-1 fluorescence intensity by 50% at the cleavage site in the control and *him-4(RNAi)* dividing germ cells. Anaphase onset = time 0. Error bars represent SEM. ***$P$ < 0.001 ($t$ test). **(C)** Quantification of the average intensity of ANI-1::GFP during membrane ingression in the control and *him-4(RNAi)* dividing germ cells. Each time point was indicated by different color scale as indicated in the timeline indicator. Error bars represent ± SEM.

established role as a scaffold protein to include a regulatory role in germline cytokinesis. HIM-4 stabilizes the cleavage membrane in the mitotic and meiotic transition zones to prevent the formation of multinucleated germ cells. This stabilization requires ANI-1. We observed that HIM-4 retains ANI-1 at the rachis bridges and that it is required for proper cytokinesis in germ cells. Upon anaphase onset, HIM-4 accumulates at the outer leaflet of the division plane membrane and moves together with the cleavage furrow as its ingresses. Depletion of HIM-4 compromised the proper recruitment of ANI-1 at the division plane, which in turn reduces the rate of contractile ring formation. Based on these results, we propose that HIM-4 regulates ANI-1 and consequently stabilizes germ cell cytokinesis, ensuring a stable syncytial gonad.

In the *C. elegans* germline, HIM-4 is mainly secreted by gonadal leader cells as a monomer that cross-links to form complexes in the rachis (Vogel & Hedgecock, 2001; Dong et al, 2006). HIM-4 tracks function as extracellular scaffolding to maintain the proper syncytial architecture. Multinucleated germ cells and enlarged rachis bridges are commonly observed in HIM-4–mutant germlines. Interestingly, similar phenotypes are documented and shown in our ANI-1 depletion results (Amini et al, 2014). In addition, depleting ANI-1 from the *him-4* mutant did not enhance the *him-4* phenotype, indicating that ANI-1 may act in the same regulatory pathway with HIM-4. In the *C. elegans* germline, ANI-1 is recruited to the rachis and forms ring-like structures at the rachis bridge (Rehain-Bell et al, 2017), and its stable rachis bridge localization is HIM-4 dependent. This suggests that HIM-4 and ANI-1 act in the same pathway. However, HIM-4 does not regulate ANI-1 function directly. This was tested by ANI-2 depletion. ANI-2 is a putative competitor of ANI-1 (Chartier et al, 2011), but depletion of ANI-2 only weakly rescued the size of the rachis bridge in *him-4*–mutant gonads, indicating HIM-4 does not promote the function of ANI-1. Furthermore, ANI-2 does not regulate the recruitment of ANI-1 to the rachis (Rehain-Bell et al, 2017), whereas both the mobility and localization of ANI-1 at the rachis bridge are compromised in the HIM-4 depletion. Thus, we suggest that HIM-4 stabilizes the localization of ANI-1 at the rachis.

This proposed idea is corroborated in dividing germ cells. Apart from localization to the rachis, HIM-4 is also present in the extracellular spaces between germ cells in the *C. elegans* gonad. Surprisingly, the immobile HIM-4 is recruited to the cleavage furrow of dividing germ cells, which is also observed in mouse embryos (Xu & Vogel, 2011b). Our time-lapse imaging clearly showed that the patches of GFP::HIM-4, which are adjacent to the cleavage site, deformed and moved towards the division plane. Although HIM-4 is highly immobile (by diffusion), HIM-4 in bulk may translocate to the division plane by mechanical force (i.e., suction effect) generated by the nascent extracellular space at the equatorial region. This recruitment of HIM-4 to the division site would further promote the recruitment and stabilization of ANI-1 to facilitate the assembly of the mature contractile ring. Therefore, HIM-4 is not required for the initiation of contractile ring formation but is required for contractile ring maturation.

Anillin is a major component in the contractile ring, which acts as a scaffolding protein to stabilize the cleavage furrow positioning during cytokinesis. Apart from its major function in cell division, anillin also has been reported to act at epithelial cell junctions (Reyes et al, 2014). Anillin localizes to epithelial cell junctions in gastrula-stage *Xenopus laevis* embryos and regulates both tight junctions and adherens junctions (Reyes et al, 2014). Similar patterning of ANI-1 and HIM-4 was observed in the *C. elegans* gonad around each of the germ cells (Fig 1B, surface view). However, we only noticed a very slight alteration of the shape of the germ cells upon HIM-4 and ANI-1 depletion, where the polygonal germ cells appeared to be a bit rounder. This might imply that HIM-4 and ANI-1 may not be required for the integrity of cell junctions in *C. elegans* germline.

How does HIM-4 regulate ANI-1 given that HIM-4 and ANI-1 are separated by a lipid bilayer? We initially considered that a transmembrane protein could likely act as intermediary by interacting with HIM-4 through the extracellular portion and to ANI-1 through the intracellular portion. However, both literature search and our screening failed to identify such candidate. The other possible mechanism might be that HIM-4 regulates the modification of the lipid composition of the germ cell membrane to promote the recruitment of ANI-1. ANI-1 has a C-terminal PH domain that has been demonstrated to be involved in PI(4,5)P2-mediated cortical recruitment (Liu et al, 2012; Sun et al, 2015). Importantly, PI(4,5)P2 is one of the key factor for furrow stability during cytokinesis (Wong et al, 2005; Logan & Mandato, 2006; Abe et al, 2012). In conclusions, determining the detailed molecular mechanism on the crosstalk between HIM-4 and ANI-1 would provide the important insight to fully understand the extracellular regulation of cytokinesis.

## Experimental procedures

### C. elegans strains
The *C. elegans* strains used in this study are listed in Table S1. All of the strains were cultured at 22°C on NGM plates seeded with *Escherichia coli* strain OP50.

### Transgenic strain construction
An N-terminal GFP knock-in transgenic strain of endogenous *ani-1* was generated using CRISPR/Cas9-triggered homologous recombination as described previously (Dickinson et al, 2015). The Cas9–sgRNA construct of *ani-1* was obtained by inserting the target sequence into pDD162, and the primers used for target insertion were ani-1-sgF (CTACACTGTAAATACAATGGGTTTTAGAGCTAGAAATAG-CAAGT) and ani-1-sgR (CAAGACATCTCGCAATAGG). A homologous repair template construct of *ani-1* was obtained via insertion of the two homology arms of *ani-1* into pDD282 digested with *ClaI* and *SpeI*, and two primer pairs were used for the construction: one pair of primers to amplify the 5′ homology arm of the *ani-1* genome insertion site (ani-1-rtF1: ACGTTGTAAAACGACGGCCAGTCGCCGGCAGTCTTGTGCTGGAG-GATG; ani-1-rtR1: TCCAGTGAACAATTCTTCTCCTTTACTCATTGTATTTA-CAGTGTAGTTCTGC) and one pair of primers to amplify the 3′ homology arm (ani-1-rtF2: CGTGATTACAAGGATGACGATGACAAGAGAATGGGGGATCA-ATTCGAT; ani-1-rtR2: TCACACAGGAAACAGCTATGACCATGTTATTGGGCAGAT-AGTTCACAT).

### RNA interference
RNAi was performed according to the feeding methods described by Timmons et al (Timmons & Fire, 1998). HT115(DE3) *E. coli* were transformed with a dsRNA expression plasmid derived from L4440

with the sequence encoding the gene of interest; these plasmids were obtained from the RNAi library. To produce RNAi-feeding plates expressing dsRNA, the bacteria were grown in LB with 100 $\mu$g/ml ampicillin overnight at 37°C and then seeded onto NGM plates containing 100 $\mu$g/ml ampicillin and 1 mM IPTG, followed by incubation at room temperature for 8 h. To deplete the two genes, the corresponding bacterial strains were mixed at a 1:1 ratio based on the optical densities of the cultures. Worms were then raised on these RNAi-feeding plates for 24–72 h before microscopy imaging.

### FRAP experiments

FRAP experiments were conducted within worms expressing GFP::ANI-1, GFP::ANI-2, NMY-2::GFP, and GFP::HIM-4 under control or *him-4*–depleted conditions. FRAP experiments were performed on an inverted spinning disk confocal microscope (Yokogawa CSU-X1) with a MicroPoint system (337 nm laser combined with 365 nm dye solution [UN1230, MP-27-365-DYE; Andor Technologies]). Photobleaching experiments were performed using the MicroPoint targeted illumination acquisition routine in MetaMorph (Molecular Devices). For photobleaching, a 60×/1.45 NA oil immersion objective was used, and the number of pulses and the MicroPoint laser power were adjusted in different samples. Time-lapse single-plane images were acquired under a 488-nm excitation laser: two frames were imaged before photobleaching, and the regions of interest (ROIs) were then photobleached. Images were acquired after photobleaching at 20-s intervals according to the parameter settings.

For data analysis, the shuffling of the time-lapse images was first adjusted with the stacks-shuffling plugin in ImageJ. Then, the intensity of the bleached region (ROI1), the non-bleached rachis region (ROI2), and the background region (ROI3) was measured with ROI manager in ImageJ. Thereafter, the obtained raw fluorescence data were normalized with easyFRAP software. In detail, the intensity of ROI1 ($I(t)_{ROI1}$) and ROI2 ($I(t)_{ROI2}$) was first normalized by subtracting the intensity of the background region ROI3 ($I(t)_{ROI2}$) using the following formula: $I(t)_{ROI1'} = I(t)_{ROI1} - I(t)_{ROI3'}$ and $I(t)_{ROI2'} = I(t)_{ROI2} - I(t)_{ROI3}$. Second, the intensity was double-normalized to correct for differences in the pre-bleaching intensity of ROI1 and differences in total fluorescence during the entire experimental procedure because of acquisition bleaching or fluctuations in laser intensity, according to the following formula:

$$I(t)' = \left\{ \frac{\frac{1}{n_{pre}}\sum_{t=1}^{n_{pre}} I(t)_{ROI2'}}{I(t)_{ROI2'}} \right\} \left\{ \frac{I(t)_{ROI1'}}{\frac{1}{n_{pre}}\sum_{t=1}^{n_{pre}} I(t)_{ROI1'}} \right\}.$$ Third, full-scale normalization was additionally corrected for differences in bleaching depth by subtracting the intensity of the first post-bleach image in ROI1 according to the following formula: $I(t)'' = \frac{I(t)' - I(t_{postbleach})'}{1 - I(t_{postbleach})}$. Finally, the normalized intensity was plotted against the elapsed time to obtain the normalized recovery curve or the mean normalized recovery curve, by averaging the intensities obtained in several experimental repeats. The sample sizes are indicated in the figures.

### Fluorescence imaging of living animals

All of the microscopy imaging experiments were performed with young adult *C. elegans* hermaphrodites. The worms were anesthetized with 0.1% tetramisole and mounted on a 3–5% agarose pad with a coverslip, which was sealed with petroleum jelly. To analyze the germ cell cytokinesis process and cytological phenotypes under gene depletion, time-lapse with Z-stack images were acquired with an inverted spinning disk confocal microscope (Yokogawa CSU-X1) under a 60×/1.4 NA objective, or using a Nikon A1R confocal microscope under a 100×/1.4 NA objective, or using a ZEISS LSM 880 microscope with an AiryScan module or with normal confocal module under a 63×/1.4 NA objective.

### Image analysis and quantification

To measure the diameter of rachis bridges, a Z-stack (interval: 0.5 $\mu$m) from the surface to the bottom layer of the germline was acquired using a Nikon A1R confocal microscope with a 100×/1.4 NA objective. For each germ cell, the clearest middle layer of the rachis bridge in the meiotic pachytene zone was used to perform measurements. The diameter of the rachis bridges was measured with ImageJ.

To measure furrow ingression, cleavage furrow fluorescence accumulation, and chromosome segregation kinetics, single central-plane time-lapsed images of NMY-2::GFP; HIM-4::GFP; GFP::ANI-1 and PH::GFP with mCherry::HIS or PH::mCherry were acquired at 10- or 20-s intervals upon anaphase onset, using an inverted spinning disk confocal microscope with a 60×/1.4 NA objective. The position of the furrow tip was manually pointed to in each frame showing the PH::GFP, PH::mCherry, NMY-2::GFP, or ANI-1::GFP signal. Furrow ingression was calculated based on the normalized cortical distance (normalized cortical distance = extent of ingression/cell width). Chromosome segregation was calculated based on the normalized chromosome segregation distance (normalized chromosome segregation distance = chromosome segregation distance/cell width). Statistical significance between different treatments or samples was determined by applying *t* test (two sample equal variance, two-tailed) using GraphPad Prism software.

## Supplementary Information

## Acknowledgements

This research was supported by Guangdong Provincial Key Laboratory of Cell Microenvironment and Disease Research (grant no. 2017B030301018), Shenzhen Key Laboratory of Cell Microenvironment (grant no. ZDSYS20140509142721429), National Natural Science Foundation of China (grant no. 31471311 and 31671409), and Shenzhen Science and Technology Innovation Commission (grant no. JCYJ20170307105005654 and JCYJ20150529152146478). We thank SUSTech Materials Characterization and Preparation Center and Life Science Research Center for providing all the equipment used in this study. We specially thank Andrew Loria for the comments on the manuscript.

### Author Contributions

H Lan: data curation, formal analysis, investigation, visualization, and writing—original draft.

X Wang: data curation, formal analysis, investigation, visualization, and writing—review and editing.

L Jiang: data curation, formal analysis, investigation, visualization, and writing—review and editing.

J Wu: data curation and investigation.

X Wan: formal analysis, investigation, visualization, and writing—review and editing.

L Zhang: data curation, formal analysis, investigation, visualization, and writing—review and editing.

D Zhang: conceptualization, data curation, funding acquisition, investigation, and writing—original draft.

Y Lin: investigation, and writing—original draft.

C Hou: writing—original draft and editing.

S Wu: funding acquisition and writing—original draft.

YC Tse: conceptualization, data curation, formal analysis, funding acquisition, investigation, visualization, project administration, and writing—original draft, review, and editing.

## Conflict of Interest Statement

The authors declare that they have no conflict of interest.

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
