## [Reviewer comments · Life Science Alliance]

Life Science Alliance

An extracellular matrix protein promotes Anillin-dependent processes in the *C. elegans* germline

Hongxia Lan, Xinyan Wang, Ling Jiang, Jianjian Wu, Xuan Wan, Lidan Zhang, Dandan Zhang, Yiyao Lin, Chunhui Hou, Shian Wu, and Yu Tse

DOI: <https://doi.org/10.26508/lsa.201800152>

Corresponding author(s): Yu Tse, Southern University of Science and Technology

Review Timeline:

Submission Date:	2018-08-09
Editorial Decision:	2018-08-10
Revision Received:	2019-03-26
Editorial Decision:	2019-04-01
Revision Received:	2019-04-04
Accepted:	2019-04-04

Scientific Editor: Andrea Leibfried

Transaction Report:

Please note that the manuscript was previously reviewed at another journal and the reports were taken into account in the decision-making process at *Life Science Alliance*. Since the original reviews are not subject to *Life Science Alliance*'s transparent review process policy, the reports and author response cannot be published.

August 10, 2018

Re: Life Science Alliance manuscript #LSA-2018-00152-T

Dr. Yu Chung Tse
Southern University of Science and Technology
No 1088, Xueyuan Rd., Xili, Nanshan District
Shenzhen 518055
China

Dear Dr. Tse,

Thank you for transferring your manuscript entitled "An extracellular matrix protein promotes Anillin-dependent processes in the *C. elegans* germline" to Life Science Alliance. The manuscript was assessed by expert reviewers at another journal before, and these reports have been transferred to us by the editor.

The reviewers who assessed your work elsewhere had split views on your work. Reviewer #1 and #2 thought that some further clarification / improvement of data representation would be needed prior to publication, while reviewer #3 and #4 found the data added during a previous round of revision too preliminary. The latter is not a concern for publication in Life Science Alliance. We would therefore like to invite you to submit a further revised version for publication in Life Science Alliance. The comments you've received from reviewer #1 and #2 should get addressed in a point-by-point response and by changes to the manuscript text and figure representation. Some data need to get re-analyzed and quantitated for this, which seems in our view straight-forward though. The major concern of reviewer #3 and #4 should get addressed by removing the currently inconclusive PIP2/ppk-1 data and associated discussion from the manuscript.

-- High-resolution figure, supplementary figure and video files uploaded as individual files: See our detailed guidelines for preparing your production-ready images, <http://life-science-alliance.org/authorguide>

B. MANUSCRIPT ORGANIZATION AND FORMATTING:

Full guidelines are available on our Instructions for Authors page, <http://life-science-alliance.org/authorguide>

Thank you for this interesting contribution to Life Science Alliance. We are looking forward to receiving your revised manuscript.

Sincerely,

April 1, 2019

RE: Life Science Alliance Manuscript #LSA-2018-00152-TR

Dr. Yu Chung Tse
Southern University of Science and Technology
1088 Xueyuan Avenue
Shenzhen 518055
China

Dear Dr. Tse,

Thank you for submitting your revised manuscript entitled "An extracellular matrix protein promotes Anillin-dependent processes in the *C. elegans* germline". We appreciate the introduced changes and would be happy to publish your paper in Life Science Alliance pending final revisions necessary to meet our formatting guidelines:

- please see the attached ms with some suggestions for text changes
- please add callouts in the text for Fig 1G and H
- we follow ICMJE authorship and author contribution guidelines, please check these and clarify.

A. FINAL FILES:

-- Summary blurb (enter in submission system): A short text summarizing in a single sentence the study (max. 200 characters including spaces). This text is used in conjunction with the titles of papers, hence should be informative and complementary to the title. It should describe the context and significance of the findings for a general readership; it should be written in the present tense

and refer to the work in the third person. Author names should not be mentioned.

B. MANUSCRIPT ORGANIZATION AND FORMATTING:

Sincerely,

April 4, 2019

RE: Life Science Alliance Manuscript #LSA-2018-00152-TRR

Dr. Yu Chung Tse
Southern University of Science and Technology
1088 Xueyuan Avenue
Shenzhen 518055
China

Dear Dr. Tse,

Thank you for submitting your Research Article entitled "An extracellular matrix protein promotes Anillin-dependent processes in the *C. elegans* germline". It is a pleasure to let you know that your manuscript is now accepted for publication in Life Science Alliance. Congratulations on this interesting work.

DISTRIBUTION OF MATERIALS:

Again, congratulations on a very nice paper. I hope you found the review process to be constructive and are pleased with how the manuscript was handled editorially. We look forward to future exciting submissions from your lab.

Sincerely,
